# Detecting and Mitigating Insertion Hallucination in Video-to-Audio Generation

## Abstract

Video-to-Audio generation has made remarkable strides in automatically synthe-
sizing sound for video. However, existing evaluation metrics, which focus on
semantic and temporal alignment, overlook a critical failure mode: models of-
ten generate acoustic events, particularly speech and music, that have no corre-
sponding visual source. We term this phenomenon Insertion Hallucination and
identify it as a systemic risk driven by dataset biases, such as the prevalence of
off-screen sounds, that remains completely undetected by current metrics. To
address this challenge, we first develop a systematic evaluation framework that
employs a majority-voting ensemble of multiple audio event detectors. We also
introduce two novel metrics to quantify the prevalence and severity of this issue:
IH@vid (the fraction of videos with hallucinations) and IH@dur (the fraction of
hallucinated duration). Building on this, we propose Posterior Feature Correction,
a novel training-free inference-time method that mitigates IH. PFC operates in a
two-pass process: it first generates an initial audio output to detect hallucinated
segments, and then regenerates the audio after masking the corresponding video
features at those timestamps. Experiments on several mainstream V2A bench-
marks first reveal that state-of-the-art models suffer from severe IH. In contrast,
our PFC method reduces both the prevalence and duration of hallucinations by
over 50% on average, without degrading, and in some cases even improving, con-
ventional metrics for audio quality and temporal synchronization. Our work is the
first to formally define, systematically measure, and effectively mitigate Insertion
Hallucination, paving the way for more reliable and faithful V2A models.

## 1 Introduction

Sound design plays a crucial role in enhancing realism and creating an immersive experience in post-
production for films, games, animations, and other multimedia content. While silent videos only
convey visual information, sound provides richer cues about temporal rhythm, spatial environment,
and emotional tone, allowing viewers to better understand and engage with the scene. For instance,
suspenseful footsteps in a horror film, distant airplane roars at an airport, or synchronized impact
sounds in action games are all indispensable elements. Traditionally, Foley effects are manually
recorded, edited, and mixed by sound designers, which is highly specialized, time-consuming, and
difficult to scale. This has motivated the development of automatic sound generation systems.

Recently, the task of Video-to-Audio (V2A) generation has attracted increasing attention as a
promising solution. V2A models aim to automatically generate sounds that are semantically rel-
evant and temporally aligned with the video content. State-of-the-art models such as MMAudio
(Cheng et al., 2025b) and ThinkSound (Liu et al., 2025) have demonstrated impressive results by
learning video-audio alignment from large-scale paired datasets. To evaluate their performance, re-
searchers typically adopt metrics such as FD-VGG (Heusel et al., 2017; Hershey et al., 2017), ISC
(Salimans et al., 2016), and DeSync (Ruder et al., 2020), which measure the semantic similarity
and temporal synchronization between generated and reference audio. These metrics have driven
remarkable progress in making V2A models produce the correct category of sounds at the correct
moments.

However, these metrics rely on the unverified assumption that generated sounds must exist in the
video. In practice, around 50% of VGGSound samples contain off-screen sounds, almost entirely

in the Speech and Music categories, while other events remain on-screen (Figure 1; Zverev et al. (2025a)). This bias predisposes models to hallucinate speech or music when visual cues are weak, yet such errors are not captured by current semantic or temporal metrics, leading to misleading assessments of model reliability.

We refer to this overlooked phenomenon as Insertion Hallucination (IH), which denotes the generation of sound events, especially speech and music, that do not exist in the video (Figure 2 illustrates an example where a model outputs speech or music despite their absence in the visual scene). Through empirical studies, we find that representative models including ThinkSound (Liu et al., 2025) and MMAudio (Cheng et al., 2025a) frequently exhibit IH on non-speech and non-music videos from mainstream datasets such as VGGSound (Chen et al., 2020) and the recently released Kling-Audio-Eval (Jun Wang, 2025). For example, a model may produce melodic music while the video only depicts sanding, or generate human voices on a vacuum cleaner video. These results suggest that IH is a systematic and widespread risk in V2A generation, yet remains largely undetected by existing metrics and has not been systematically addressed in prior work.

To systematically investigate IH, we develop an end-to-end evaluation framework. We begin with an automatic detection pipeline that identifies hallucinated speech and music segments by integrating three audio event detectors: inaSpeechSegmenter (Doukhan et al., 2018), YAMNet (Ellis et al., 2019), and PANNs (Kong et al., 2020), and fusing their outputs with majority voting. The pipeline is then validated on a human-annotated set to assess accuracy. Finally, we introduce two metrics: IH@vid (the fraction of videos containing hallucination) and IH@dur (the fraction of hallucinated duration).

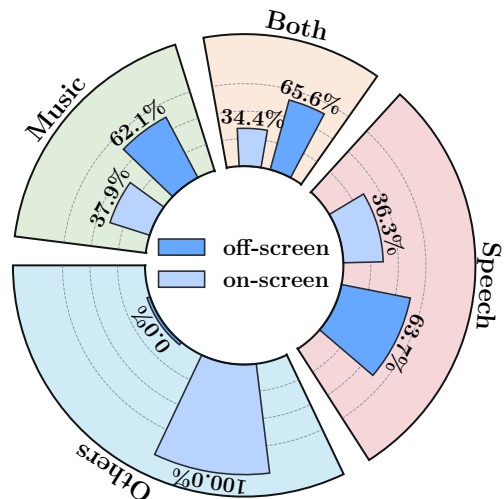

Figure 1: Distribution of on-screen and off-screen sounds in VGGSound. Around 50% of all samples contain off-screen sounds, which are concentrated in the Speech and Music categories, while other events are almost entirely on-screen. This highlights a dataset bias relevant to hallucination analysis.

Building on this, we propose a novel inference-time correction method called Posterior Feature Correction (PFC). PFC does not require retraining the model. Instead, it runs a two-pass generation process. In the first pass, the model generates audio and we detect hallucinated segments. In the second pass, we mask the video features at those segments by replacing them with empty features, and regenerate the audio. This forces the model to rely on contextual or label information instead of unreliable visual features, preventing it from degenerating into speech or music hallucinations. Experiments show that PFC significantly reduces IH@vid and IH@dur while preserving standard metrics such as FD-VGG, ISC, and DeSync.

Our main contributions are summarized as follows:

- We are the first to define Insertion Hallucination (IH) in audio generation, revealing realism as a critical risk dimension that is completely overlooked by existing evaluation metrics.

- We build an IH evaluation framework combining multi-detector voting and human verification, and propose two metrics (IH@vid and IH@dur) to quantify models' hallucination tendency.

- We propose Posterior Feature Correction (PFC), a training-free inference-time method that significantly reduces IH while maintaining conventional metrics, and demonstrate its effectiveness on multiple V2A benchmarks.

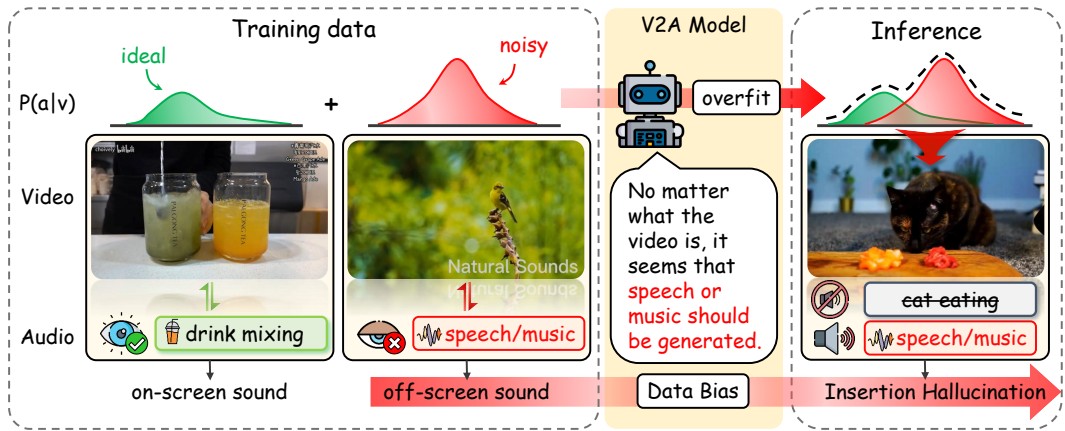

Figure 2: Example of Insertion Hallucination (IH) in video-to-audio generation. Training data often include off-screen speech or music, which biases models to reproduce them. As a result, during inference the model may generate speech or music even when the video only depicts other events.

## 2    RELATED WORK

### 2.1    VIDEO-TO-AUDIO GENERATION

Early work on Video-to-Audio generation was dominated by Generative Adversarial Networks (Chatterjee & Cherian, 2020; Pascual et al., 2017; Ferreira et al., 2022), but recent advances have shifted toward models that produce higher-quality audio with stronger audio-visual alignment. A key direction is improving representation alignment: Diff-Foley (Luo et al., 2023) employs contrastive audio-visual pretraining to learn temporally and semantically aligned features that guide a latent diffusion model, achieving substantial gains in synchronization and relevance.

With the rise of more capable generative models, research has expanded toward controllability and practicality. FoleyCrafter (Zhang et al., 2024) adapts a pre-trained text-to-audio model with semantic and temporal controllers, enabling prompt-based control with precise alignment. Data expansion is another path: MMAudio (Cheng et al., 2025a) unifies video-audio and large-scale text-audio data for richer semantics, while MultiFoley (Chen et al., 2025) conditions on both text and audio for flexible user guidance and high-fidelity synchronization.

The frontier is now moving beyond direct mapping to incorporate reasoning. ThinkSound (Liu et al., 2025) introduces a Chain-of-Thought framework in which a multimodal large language model produces interpretable reasoning steps that guide audio generation, transforming the task into a cognitively driven process.

Nevertheless, evaluation remains centered on semantic relevance and synchronization, neglecting whether generated sounds should appear in the video at all. Current metrics cannot capture hallucinations such as spurious speech or music, leaving a critical gap that our work aims to fill.

### 2.2    HALLUCINATION IN LARGE LANGUAGE MODELS

Hallucination is a core challenge for large-scale AI models, with related research expanding from Large Language Models to the multimodal domain. In Large Language Models, researchers mitigate insertion hallucination by introducing external knowledge bases (Lee et al., 2022) or enhancing internal consistency (Mündler et al., 2023). This issue manifests in Vision-Language Models as object hallucination, where a model describes non-existent objects in an image. The academic community has established dedicated evaluation benchmarks such as POPE (Yifan Li & Wen, 2023) and proposed solutions such as Object-Aware Preference Optimization (Chen et al., 2024b; Compagnoni et al., 2025). Recently, the evaluation of hallucination has extended to the audio-visual domain; for instance, AVHBench (Sung-Bin et al., 2024) designs cross-modal understanding tasks to assess whether a model exhibits audio-driven or video-driven hallucinations. However, while these studies prove that hallucination is a common risk in multimodal models, the evaluation endpoint of all

existing work, whether for language, vision, or audio-visuals, is exclusively focused on whether the generated textual output contains hallucinations. The phenomenon of the audio itself being the subject of hallucination, such as a Video-to-Audio model generating sound that contradicts the visual scene, remains an unexplored research gap.

## 2.3 OFF-SCREEN SOUND GENERATION

A notable recent trend in the V2A field is the research on generating off-screen sound. Many researchers have observed that existing video datasets commonly contain off-screen audio events, and they aim to make models learn and align with this characteristic to generate more complete and immersive holistic soundscapes. For instance, VinTAGe (Kushwaha & Tian, 2025) leverages additional information such as text to assist in generating off-screen sounds, while Action2Sound (Chen et al., 2024a) independently models off-screen ambient audio by separating it from foreground sounds. The importance of this trend is also reflected in the evolution of evaluation methods: VGGSounder (Zverev et al., 2025b) was the first to introduce an off-screen sound dimension into its evaluation framework. By comparing model performance with and without visual cues, it revealed a common "over-reliance on vision" bias in existing models, thereby emphasizing the importance of independent audio understanding capabilities.

However, we argue that pursuing alignment with off-screen sounds poses a risk, as it may sacrifice the model's fidelity to the visual content and its generalization capabilities. In contrast, we advocate for the "What You See Is What You Get" principle. We believe that a model should first focus on generating faithful and reliable audio for visible visual content, as this is the fundamental basis for building controllable and trustworthy generative models.

## 3 METHODOLOGY FOR MEASURING INSERTION HALLUCINATION

### 3.1 PROBLEM ANALYSIS AND DEFINITION

Video-to-Audio generation models learn a conditional mapping $P(a|v)$ from visual input $v$ to audio output $a$ using paired training data. However, mainstream datasets contain a high prevalence of off-screen sounds, particularly speech and music, which introduces a systematic bias. When visual cues are weak or ambiguous, models often default to reproducing these frequent patterns rather than faithfully rendering scene-consistent audio.

We define this failure mode as *Insertion Hallucination* (IH): the generation of structured acoustic events that have no corresponding source in the visual content. While IH could in principle include any spurious sound, we focus on speech and music for three reasons: (1) they are the most frequent off-screen sounds in mainstream corpora, with over half of VGGSound samples exhibiting this bias (Figure 1); (2) they are perceptually salient events whose presence can strongly disrupt immersion; and (3) mature detection tools are available, enabling reliable identification.

Formally, given a video–label pair $(v, y)$, where $y$ specifies the ground-truth sound category, and an audio prediction $\hat{a} = G(v)$ from a model $G$, we define the hallucination indicator as:

$$\text{is\_IH}(v, y, \hat{a}) = \begin{cases} 1, & \text{if } y \notin \mathcal{Y}_{sm} \text{ and } D(\hat{a}) \neq \emptyset, \\ 0, & \text{otherwise,} \end{cases}$$

where $\mathcal{Y}_{sm}$ is the set of speech and music labels and $D(\hat{a})$ denotes detected hallucinated segments.

### 3.2 MULTI-DETECTOR ENSEMBLE FRAMEWORK

Detecting hallucinations reliably requires addressing the limitations of individual audio classifiers. To this end, we design a multi-detector ensemble that combines three complementary detectors: inaSpeechSegmenter (Doukhan et al., 2018), YAMNet (Ellis et al., 2019), and PANNs (Kong et al., 2020).

Our pipeline consists of three stages:

1. **Candidate Filtering.** Samples with ground-truth labels in $\mathcal{Y}_{sm}$ are excluded, ensuring that evaluation is limited to videos where speech and music are not expected.

2. **Multi-Detector Analysis.** Each detector independently identifies speech and music segments based on its model-specific decision boundary.

3. **Ensemble Fusion.** Final predictions are obtained by majority voting across detectors, balancing precision and recall while reducing detector-specific biases:

$$D_R(\hat{a}) = \big\{ s \,\big|\, \sum_{k=1}^{K} \mathbf{1}[s \in D_k(\hat{a})] \geq \lceil K/2 \rceil \big\}.$$

This ensemble balances precision and recall while being robust to detector-specific biases. We validate its reliability against human annotations (Section 5).

### 3.3 EVALUATION METRICS

To quantify hallucination behavior, we introduce two complementary metrics. Let $M$ denote the number of evaluated samples, $d_i$ the total hallucinated duration of sample $i$, and $T_i$ its total length.

$$\text{IH@vid} = \frac{1}{M} \sum_{i=1}^{M} \mathbf{1}[d_i > 0], \qquad \text{IH@dur} = \frac{1}{M} \sum_{i=1}^{M} \frac{d_i}{T_i}.$$

IH@vid measures the proportion of audios that contain hallucination (prevalence), while IH@dur measures the proportion of hallucinated duration (severity).

## 4 POSTERIOR FEATURE CORRECTION

Despite recent advances, state-of-the-art V2A models still exhibit Insertion Hallucination when visual inputs provide insufficient cues for reliable generation. This failure mode reflects a systematic reliance on strong dataset priors, in particular speech and music, whenever the visual signal is ambiguous. To address this issue, we propose *Posterior Feature Correction* (PFC), a training-free inference method that dynamically masks unreliable video features identified through hallucination detection.

### 4.1 METHOD MOTIVATION

We observe that hallucinations arise most often when visual representations fail to provide discriminative guidance for audio generation. This suggests a feedback mechanism: if we can identify where hallucinations occur, we can infer where visual features are unreliable.

The key insight is that V2A models exhibit predictable failure modes. When visual encoders produce ambiguous representations, for example due to visual similarity between acoustically different events, poor lighting, or out-of-distribution content, models fall back to generating high-frequency training patterns. The location of these hallucinations thus serves as a diagnostic signal for visual uncertainty.

We propose Posterior Feature Correction (PFC), which exploits this signal through a two-stage process: first generate audio to identify problematic regions, then regenerate with visual features masked at those locations. By removing unreliable visual cues, we force the model to rely on more conservative generation strategies and stronger contextual information.

This approach is inspired by self-correction mechanisms in other domains (Madaan et al., 2023; Shinn et al., 2023; Huang et al., 2023), but uniquely leverages the temporal structure of audio-visual alignment for targeted feature intervention. An overview of the process is shown in Figure 3.

### 4.2 ALGORITHM DESIGN

PFC operates in two inference passes:

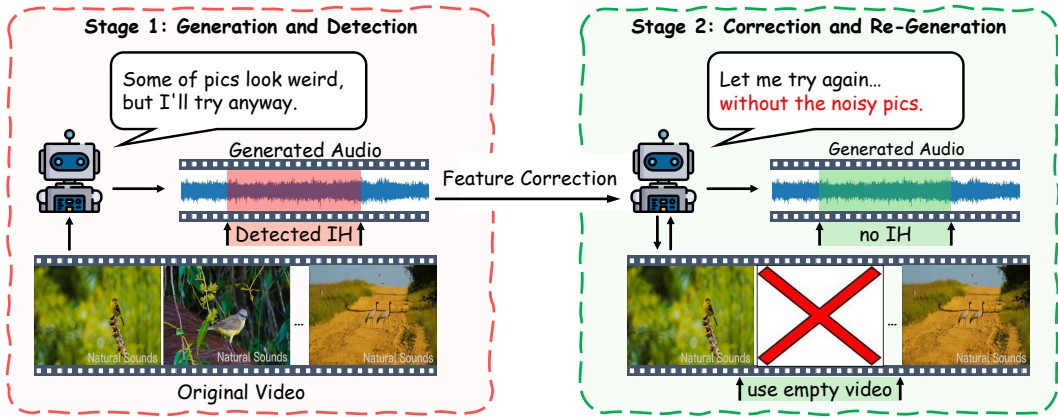

Figure 3: Overview of *Posterior Feature Correction* (PFC). The method first detects hallucination segments (red) in the initial generation, then masks corresponding video features with a learnable empty token to obtain a corrected output (green).

**Stage 1: Detection.** Given an input video $v$ with visual features $f_v$, the model generates a preliminary audio clip $\hat{a} = G(v, f_v)$. We then apply our hallucination detector $D(\cdot)$ to obtain a set of hallucination intervals $\mathcal{H} = D(\hat{a})$, where each $\tau = [s, e] \in \mathcal{H}$ marks a time span predicted as speech or music hallucination.

**Stage 2: Correction.** We construct corrected features $f'_v$ by replacing the features at hallucinated timestamps with an empty representation $\emptyset_v$, a special empty token provided by the pretrained model. This design preserves temporal structure while suppressing misleading cues:

$$f'_v(t) = \begin{cases} \emptyset_v, & t \in \bigcup_{\tau \in \mathcal{H}} \tau \\ f_v(t), & \text{otherwise.} \end{cases}$$

The corrected features $f'_v$ are then fed back to the same model, yielding a revised output $\hat{a}' = G(v, f'_v)$.

This two-stage process exploits hallucination locations as uncertainty indicators and intervenes only where necessary. By removing unreliable cues, the model is forced to rely on contextual information and more conservative generation strategies, thereby reducing IH while preserving semantic accuracy and synchronization elsewhere.

## 5 EXPERIMENTS

This section presents a comprehensive empirical validation of our proposed framework. We first validate our IH detection pipeline on a human-annotated dataset (Section 5.2). We then apply it to assess the prevalence of Insertion Hallucination (IH) in state-of-the-art models and evaluate our Posterior Feature Correction (PFC) method (Section 5.3). Finally, we analyze PFC's core components via an ablation study (Section 5.4) and compare it against alternative correction methods (Section 5.5).

### 5.1 EXPERIMENTAL SETUP

We validated our IH detection pipeline using three audio event detectors, inaSpeechSegmenter, YAMNet, and PANNs, on the human-annotated set described in Appendix 7.1, measuring Precision, Recall, $F_\beta$-score, and IoU. We then applied the validated metrics to evaluate two representative V2A systems: MMAudio (Cheng et al., 2025a), a multimodal framework with synchronization and flow-matching generation, and ThinkSound (Liu et al., 2025), a reasoning-based model that uses Chain-of-Thought for visual and temporal modeling. Experiments were conducted on three benchmarks: Kling-Audio-Eval (Jun Wang, 2025) (20k clips, 1.9k classes, stricter filtering of off-screen sounds), VGGSound (Chen et al., 2020) (200k clips, 310 classes, known off-screen bias),

and AVE (Tian et al., 2018) (4,143 clips, 28 classes, frame-level annotations). Evaluation covered hallucination-specific metrics (IH@vid, IH@dur), distributional metrics ($FD_{PANNs}$, $KL_{PANNs}$), semantic/quality metrics (ISC, IB-score), and temporal alignment (DeSync).

## 5.2 VALIDATION OF IH METRICS

We validated the reliability of our IH detection pipeline on a dedicated human-annotated dataset, where clips were manually labeled by consensus for the presence of speech and music hallucinations (see Appendix 7.1). However, the task of defining precise temporal boundaries for acoustic events is inherently subjective and prone to ambiguity. This makes evaluating a detector solely on its ability to perfectly replicate these boundaries (i.e., maximizing recall) a potentially misleading measure of performance. We contend that a more robust criterion for reliability is ensuring that any detected segment corresponds to a genuine, human-verified event, which places a stronger emphasis on **precision**.

To formally reflect this principle in our evaluation, we adopt the generalized $F_\beta$-score, defined as:

$$F_\beta = (1 + \beta^2) \cdot \frac{\text{Precision} \cdot \text{Recall}}{\beta^2 \cdot \text{Precision} + \text{Recall}}$$

We specifically set $\beta = 0.5$, which gives twice the weight to precision over recall, thereby aligning our quantitative evaluation with the goal of ensuring high detector reliability.

Table 1 reports the performance of our detection methods. Notably, the individual **PANNs** model achieves the highest $F_{0.5}$ score (82.97). However, the **Majority Vote (MV)** ensemble attains a nearly identical score (82.62) while providing the added robustness of a multi-detector consensus. We therefore select **MV** for all subsequent experiments, concluding that the critical benefit of ensemble robustness outweighs the negligible difference in performance scores.

Table 1: Performance of individual detectors and fusion strategies on the human-annotated validation set, using the $F_{0.5}$ score to emphasize precision. Best results are in bold, second-best are underlined.

| Method | Precision | Recall | $F_{0.5}$ | IoU |
|---|---|---|---|---|
| **Individual Detectors** | | | | |
| inaSpeechSegmenter | 79.53 | 68.18 | 76.97 | 58.00 |
| YAMNet | 79.15 | 74.54 | 78.18 | 62.30 |
| PANNs | 85.40 | 74.51 | **82.97** | 66.09 |
| **Fusion Strategies** | | | | |
| AND | **90.93** | 55.45 | 80.61 | 52.54 |
| OR | 73.68 | **87.32** | 76.06 | **66.56** |
| MV | 84.94 | 74.49 | 82.62 | 65.80 |

## 5.3 INSERTION HALLUCINATION ASSESSMENT

We next apply our validated IH metrics to state-of-the-art V2A models and to evaluate the effectiveness of our proposed PFC method.

**Baseline models exhibit systematic hallucination.** Table 2 reports results across Kling-Audio-Eval, VGGSound, and AVE. Both MMAudio and ThinkSound generate hallucinations in a substantial portion of videos (IH@vid 12–24%), with spurious speech or music often occupying 4–15% of the total duration. These findings establish that IH is not a rare anomaly but a widespread failure pattern in current V2A systems. The Ground-truth (GT) row, obtained by running our proposed IH detection pipeline on the dataset's reference audio, shows small non-zero IH values that reflect unavoidable dataset biases such as residual off-screen sounds or loosely aligned labels.

**Posterior Feature Correction substantially reduces hallucination.** Across all benchmarks, PFC consistently lowers hallucination rates. The effect is strongest on the more diverse Kling-Audio-Eval and AVE datasets, where IH@vid and IH@dur drop by 40 to 65%. On VGGSound, the main

Table 2: Results on Kling-Audio-Eval, VGGSound, and AVE, showing that PFC consistently reduces hallucinations without degrading quality or synchronization.

| | IH@vid ↓ | IH@dur ↓ | FD ↓ | KL ↓ | ISC ↑ | IB ↑ | DeSync ↓ |
|---|---|---|---|---|---|---|---|
| **Kling-Audio-Eval** | | | | | | | |
| GT | 5.23 | 1.42 | – | – | – | – | – |
| mmaudio | 12.91 | 4.55 | 10.476 | 2.502 | 8.342 | 0.3425 | 0.6102 |
| + PFC | 6.14 | 2.45 | 10.964 | 2.457 | 8.231 | 0.3420 | 0.6070 |
| Δ | **52.4%** | **46.2%** | 4.7% | 1.8% | 1.3% | 0.1% | 0.5% |
| thinksound | 24.34 | 14.69 | 12.478 | 2.760 | 5.49 | 0.2067 | 0.7366 |
| + PFC | 9.24 | 5.18 | 12.352 | 2.528 | 5.431 | 0.2174 | 0.7064 |
| Δ | **62.0%** | **64.7%** | 1.0% | 8.4% | 1.1% | 5.2% | 4.1% |
| **VGGSound** | | | | | | | |
| GT | 11.00 | 2.60 | – | – | – | – | – |
| mmaudio | 16.33 | 6.09 | 6.870 | 1.813 | 7.008 | 0.3391 | 0.6010 |
| + PFC | 8.91 | 5.47 | 6.489 | 1.780 | 7.127 | 0.3384 | 0.5941 |
| Δ | **45.4%** | **10.2%** | 5.5% | 1.8% | 1.7% | 0.2% | 1.1% |
| thinksound | 13.04 | 5.17 | 6.666 | 2.015 | 5.726 | 0.2263 | 0.7167 |
| + PFC | 6.28 | 3.86 | 6.568 | 1.953 | 5.797 | 0.2242 | 0.7240 |
| Δ | **51.9%** | **25.4%** | 1.5% | 3.1% | 1.2% | 0.9% | 1.0% |
| **AVE** | | | | | | | |
| GT | 15.29 | 1.57 | – | – | – | – | – |
| mmaudio | 13.02 | 3.07 | 3.209 | 1.473 | 6.485 | 0.3776 | 0.5460 |
| + PFC | 6.05 | 1.82 | 3.244 | 1.462 | 6.517 | 0.3771 | 0.5619 |
| Δ | **53.5%** | **40.7%** | 1.1% | 0.7% | 0.5% | 0.1% | 2.9% |
| thinksound | 19.07 | 7.40 | 8.231 | 1.921 | 5.430 | 0.2500 | 0.7195 |
| + PFC | 10.23 | 3.02 | 7.385 | 1.946 | 5.358 | 0.2505 | 0.7181 |
| Δ | **46.4%** | **59.2%** | 10.3% | 1.3% | 1.3% | 0.4% | 0.2% |

training-domain dataset, PFC still reduces hallucination frequency by 45 to 52%, but the reduction in duration is smaller (10 to 25%). This indicates that models overfit to training biases, making in-domain hallucinations harder to suppress, and highlights PFC's strength in improving generalization to out-of-domain data.

**Conventional metrics remain robust, showing no systematic degradation.** Crucially, this targeted reduction in hallucination does not come at the cost of overall generation quality. As shown in the Δ rows, conventional metrics such as FD, KL, and DeSync exhibit only minor fluctuations, with most changes falling below 3%. We even observe several instances of notable improvement, such as a 10.3% enhancement in FD and an 8.4% gain in KL for ThinkSound, suggesting that removing misleading visual features can sometimes help the model produce higher-quality audio. This stability confirms that PFC is a non-destructive method that precisely targets unwanted content without degrading audio quality, diversity, or temporal alignment.

## 5.4 ABLATION STUDY OF REPLACEMENT STRATEGIES

To validate the effectiveness of our proposed feature correction strategy, we conduct an ablation study to demonstrate that precisely targeting and correcting "problematic" visual features is crucial. We compare our method (+ PFC) against two non-precise replacement strategies: random replacement (+ Random) and complement replacement (+ ∼PFC), which corrects the non-hallucinated segments identified by our detector.

The results are shown in Figure 4 (Left). Compared with the baseline, both non-precise strategies yield only limited or inconsistent improvements. In particular, the complement replacement (+∼PFC) strategy performs even worse than random replacement on the AVE dataset, indicating

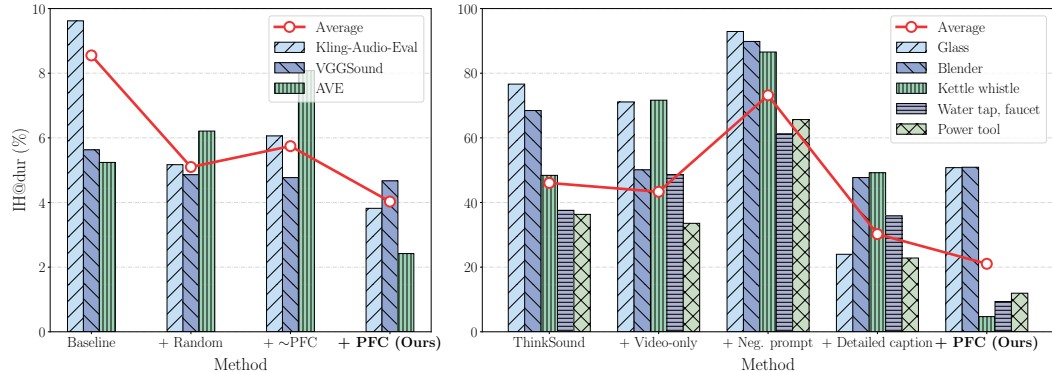

Figure 4: (Left) Ablation on feature correction, showing PFC consistently achieves the lowest IH@dur across datasets. (Right) Comparison with alternative correction methods on top-5 sublabels; PFC attains the best average IH@dur while detailed captions can outperform on object-centric classes.

that modifying non-problematic regions is not only unhelpful but can also be harmful. This confirms that the segments identified by our detector are indeed the critical ones to correct.

In contrast, our proposed PFC method consistently achieves the lowest IH@dur across all datasets, with average performance substantially surpassing all alternatives. These findings demonstrate that PFC's effectiveness comes from its ability to precisely identify and correct the video segments most likely to mislead the model, rather than relying on arbitrary or random replacement.

## 5.5 Comparison with Alternative Correction Methods

Beyond ablating feature replacement strategies, we further compare PFC with alternative correction methods that exploit different modalities. To ensure robustness, we evaluate on the five Kling-Audio-Eval sublabels where the baseline exhibits the highest hallucination rates.

Results in Figure 4 (Right) show that PFC again achieves the best average performance, attaining the lowest mean IH@dur among all methods. This highlights its effectiveness even on the most challenging cases.

However, PFC is not universally superior. For object-centric categories such as *Glass* and *Blender*, detailed captions outperform PFC in reducing hallucinations, suggesting that fine-grained textual descriptions provide stronger semantic constraints than feature correction alone.

This complementary behavior points to a promising avenue for future research: combining input-level textual guidance (e.g., detailed captions or prompts) with our posterior, feature-level correction (PFC) could potentially yield an even more powerful and reliable method for suppressing hallucinations across diverse categories.

## 6 Conclusion

We studied a neglected failure mode in video-to-audio generation: *Insertion Hallucination* (IH), where models synthesize speech or music that does not exist in the video. We introduced an evaluation framework that explicitly measures authenticity via a validated multi-detector ensemble and two metrics, IH@vid and IH@dur. On three benchmarks, these metrics reveal that state-of-the-art systems frequently hallucinate, even when conventional semantic and temporal scores appear strong.

To mitigate IH, we proposed *Posterior Feature Correction* (PFC), a training-free, two-pass procedure that detects hallucinated segments and masks the corresponding video features with a learned empty token in a second pass. PFC consistently reduces both the prevalence and duration of hallucinations across datasets while preserving (and sometimes improving) standard distribution, semantic, and synchronization metrics.

ETHICS STATEMENT

All experiments in this paper are conducted on publicly available datasets (VGGSound and Kling-Audio-Eval). The human-annotated validation set was created solely for evaluation, with no personally identifiable or sensitive information involved. Annotators participated voluntarily and were fairly compensated. Our method aims to improve model reliability by reducing hallucination, but we acknowledge that generative audio may still pose risks of misuse, which require broader safeguards beyond the scope of this work.

REPRODUCIBILITY STATEMENT

We describe datasets, models, detectors, and evaluation metrics in detail in Section 5. Appendix 7.1 provides the annotation pipeline. All code, configuration files, and annotated evaluation splits will be released to enable full reproduction of our results.

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

# 7 APPENDIX

## 7.1 HUMAN ANNOTATION PIPELINE

**Sampling.** We constructed a human-annotated validation set from the outputs of two representative V2A systems, MMAudio and ThinkSound, on the Kling-Audio-Eval benchmark. For each sublabel in Kling-Audio-Eval, we randomly selected 20 generated audio clips per model, yielding over 900 clips in total and more than 9,000 seconds of audio.[1] This per-sublabel sampling ensures broad coverage of object- and scene-centric categories while keeping the annotation workload tractable.

**Annotation Interface.** We implemented a web-based tool to support fine-grained temporal labeling (Figure 5). Annotators could (i) view the paired video, (ii) inspect a mel-spectrogram of the generated audio, (iii) scrub or play arbitrary time spans by dragging on the timeline, and (iv) add multiple segment labels of type speech or music per clip. Timestamps were recorded in seconds with a resolution of $0.01\,\text{s}$; the UI exposed zoom controls to facilitate precise boundary placement. The interface enforced basic constraints (start ¡ end, no negative durations) and warned on overlapping segments of the same type. All edits (add, move, delete) were versioned locally and only finalized upon explicit submission.

**Labeling Guidelines (What to mark).** Annotators were instructed to mark a segment if and only if the generated audio contains audible speech (spoken voice, narration, conversation, shouting, etc.) or music (melodic or harmonic content, instrumental or vocal), and such content does not have a plausible source in the visible scene. Ambiguous non-speech vocalizations (e.g., coughing, humming) were not labeled unless they clearly constitute speech or singing. Background environmental noise (wind, crowd murmur without intelligible speech, mechanical hums) and percussive impacts that do not form music should not be labeled. Segments shorter than $0.2\,\text{s}$ were discouraged unless clearly perceptible; short gaps $< 0.15\,\text{s}$ between adjacent labels of the same type could be left un-split and were merged during post-processing.

**Boundary Conventions (How to mark).** Boundaries were aligned to the earliest perceptual onset and latest perceptual offset of the target content at $0.01\,\text{s}$ granularity. To reduce over-fragmentation, adjacent labels of the same type separated by gaps $< 0.15\,\text{s}$ were merged in aggregation (Section 7.1). During metric computation, we applied a symmetric tolerance of $0.05\,\text{s}$ around annotated boundaries when matching against detector outputs.

---

[1]Counts refer to post-filtered items that passed loading and playback checks.

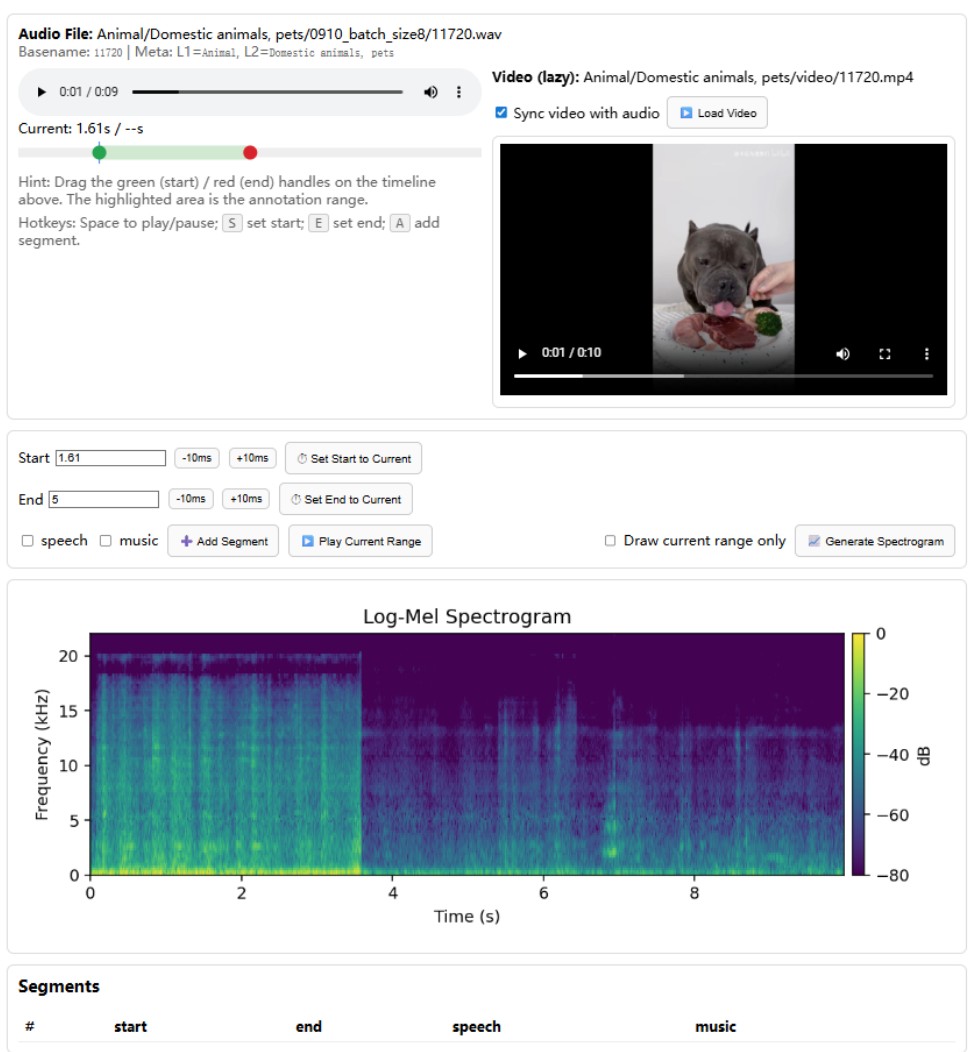

Figure 5: Web interface for human annotation. Annotators can view the paired video and the spectrogram, select arbitrary time spans for playback, and mark multiple speech or music segments per clip with 0.01 s precision.

**Annotator Instructions and Roles.** Two primary annotators independently labeled every clip; a third adjudicator resolved disagreements. Annotators worked with over-ear headphones in a quiet environment and were asked to keep the playback volume consistent across sessions. Each session began with a brief calibration: annotators labeled five warm-up clips and discussed edge cases with the adjudicator using the written guidelines.

**Quality Control.** We performed three checks before accepting a submission: (i) schema checks: well-formed JSON/CSV, valid types, monotone timestamps; (ii) consistency checks: no duplicate segments, no impossible overlaps of identical types after the merge rule; (iii) cross-annotator comparison: clip-level flags (any hallucination vs. none) and segment-level overlaps using an IoU criterion. If primary annotators disagreed on the clip-level decision or if segment alignment IoU was low, the adjudicator reviewed the audio and video and issued a final consensus label set (kept as a separate adjudicated split).

**Aggregation and Export.** We exported per-clip annotations as line records with fields: clip_id, model, sublabel, segment_type (speech or music), start, end. The fields start and end are in seconds rounded to 0.01. Post-processing merged same-type segments separated by gaps $< 0.15$ s and re-

moved residual fragments $< 0.2\,$s unless the adjudicator explicitly kept them (rare but allowed for clearly audible bursts). All timestamps remained in the clip's original time base (no resampling of the reference clock).

**Statistics.** The final set contains over 1,000 clips (more than 10,000 seconds in total). Multiple segments per clip are common; speech and music occurrences are uneven across sublabels, reflecting category-specific priors. We report clip-level hallucination prevalence (any segment present) and average hallucinated duration per clip in Section 5.2; per-category breakdowns are provided in the supplementary material.

**Intended Use.** This human-annotated set is used exclusively to validate the reliability of the IH detection pipeline (Appendix 7.1 and Section 5.2). It is not used for model training or fine-tuning, nor to design PFC. All figures reported in the main paper that depend on human labels reference the adjudicated split.

