# OpenReview forum: "Detecting and Mitigating Insertion Hallucination in Video-to-Audio Generation"
_ICLR.cc/2026/Conference — ICLR 2026 Conference Withdrawn Submission_

### Official Review · Reviewer_7uXK · 2025-10-26

**Soundness:** 2
**Presentation:** 3
**Contribution:** 2
**Rating:** 2
**Confidence:** 4

**Summary:**

This paper addresses "Insertion Hallucination" in Video-to-Audio (V2A) generation, where models generate sounds like speech or music without a visual source. The authors propose a framework to measure this issue using an ensemble of existing audio detectors and introduce two metrics, IH@vid and IH@dur. To mitigate the problem, they present Posterior Feature Correction (PFC), a training-free, two-pass inference method. PFC first generates audio to find hallucinated segments, then masks the corresponding video features and regenerates the audio. Experiments show PFC reduces hallucinations by over 50% on average across several benchmarks. The work claims to be the first to systematically define, measure, and mitigate this failure mode in V2A models.

**Strengths:**

1. The paper effectively scopes a known issue (hallucination from bias) within the V2A domain and provides a practical way to measure it.

2. PFC is a simple, training-free method that works. This makes it a useful and accessible baseline for future work.

3. The experiments are comprehensive. The authors test their claims across multiple state-of-the-art models and standard datasets, and the inclusion of human validation for the detection pipeline is a strength.

**Weaknesses:**

1. The paper has limited conceptual novelty. The contribution is best described as a solid piece of engineering and system-building, applying established concepts to a new domain. It defines a problem, builds a measurement tool from existing parts, and applies a known correction strategy. However, it does not introduce new fundamental techniques.

2. The PFC method doubles inference time and computational cost by design. This is a major practical drawback for any real-world application. The paper fails to discuss or quantify this trade-off.

3. The method only targets speech and music and is entirely dependent on the performance of external, pre-trained detectors. It is not a general solution for audio hallucination. A failure or limitation in the detectors directly translates to a failure in PFC.

4. The term "special empty token" has been vaguely used for masking, which is a critical implementation detail.

**Questions:**

1. The two-pass PFC method doubles inference cost. Please analyze and discuss about this significant trade-off. Is there a viable path toward a single-pass solution?

2. The core ideas (identifying a problem from dataset bias, ensembling off-the-shelf classifiers for detection, and using a two-pass self-correction mechanism) are applications of established concepts. Could the authors suggest what they see as the core conceptual novelty of this work, beyond applying these ideas to the V2A domain for the first time?

3. The paper states PFC uses a "special empty token provided by the pretrained model". What exactly is this token?

---

### Official Review · Reviewer_Mzcw · 2025-10-31

**Soundness:** 2
**Presentation:** 3
**Contribution:** 2
**Rating:** 4
**Confidence:** 4

**Summary:**

This paper investigates the issue of off-screen sound hallucination, where models trained on videos containing off-screen sound mistakenly associate such audio with the visible video content, leading to hallucinations during inference. After identifying this risk, the paper argues that existing evaluation metrics fail to detect this type of hallucination and proposes two new metrics designed to capture insertion hallucinations. In addition, the paper introduces an inference-time correction technique capable of identifying and mitigating such hallucinations during testing.

**Strengths:**

1. The mitigation of the insertion hallucination makes sense.

2. The paper writing is clear.

3. The performance gain especially on reducing insertion hallucination with the proposed correction technique is significant.

**Weaknesses:**

1. The robustness of the correction technique is a key concern. For example, replacing hallucinated clips with empty representations may alter the feature distribution, potentially introducing out-of-distribution (OOD) samples.

2. The detection and mitigation components are closely coupled. It would be important to analyze how the model performs when the detection accuracy decreases.

3. Some methodological details are missing. In particular, the hallucination detectors play a critical role in the proposed approach, yet this component is not thoroughly examined in either the method description or the experimental analysis.

**Questions:**

1. After detection of the hallucination, the correction is performed in a hard-threshold way, where the feature is replaced with an empty representation. In this case, how to guarantee smoothness of the generated audio?

2. Meanwhile, what if the video clip with hallucination also contain some on-screen sound, how does the model distinguish such on-screen sound and the off-screen sound, to make sure that only those off-screen sound are removed.

3. It's not clear how those three audio event detectors serve as the hallucination detectors. Detailed explanations are needed, as it's one critical component of the proposed correction pipeline.

4.The mitigation strongly depends on the detection of hallucination. What if the detection is not reliable? How does the model perform in such case?

---

### Official Review · Reviewer_i4a5 · 2025-11-01

**Soundness:** 3
**Presentation:** 3
**Contribution:** 2
**Rating:** 2
**Confidence:** 4

**Summary:**

This paper identifies a critical failure mode in video-to-audio (V2A) generation termed Insertion Hallucination (IH), where models generate non-existent speech or music. This problem stems from dataset biases, as training data often contains off-screen sounds, and remains undetected by conventional evaluation metrics. To quantify IH, the authors introduce a detection framework using an ensemble of audio event classifiers and propose two metrics: IH@vid and IH@dur.

Experiments reveal that state-of-the-art V2A models suffer from pervasive IH. To mitigate it, the authors propose Posterior Feature Correction (PFC), a training-free method that first detects hallucinated segments and then regenerates the audio with corresponding video features masked. This two-step process effectively reduces unreliable visual cues that trigger hallucinations.

Results show that PFC reduces the prevalence and duration of hallucinations by over 50% on average across multiple benchmarks. Crucially, it achieves this without degrading standard audio quality and synchronization metrics. This work is the first to systematically define, measure, and mitigate insertion hallucinations, enhancing the reliability of V2A models.

**Strengths:**

* Novel Problem Identification:
The paper is the first to formally define and systematically study Insertion Hallucination (IH) in video-to-audio generation — a critical but previously overlooked failure mode where models generate sounds (especially speech or music) not present in the visual scene. This addresses a significant gap in the evaluation of V2A models.
* Rigorous Evaluation Framework:
The authors develop a pipeline based on a majority-voting ensemble of multiple audio event detectors. They also introduce two intuitive and well-motivated metrics — IH@vid and IH@dur — to quantify both the prevalence and severity of hallucinations.
* Effective and Practical Mitigation Method:
The proposed Posterior Feature Correction (PFC) is a training-free method that significantly reduces hallucinations without degrading conventional audio quality or synchronization metrics. Its two-pass design is both simple and empirically effective.

**Weaknesses:**

* Limited Scope of Hallucination Analysis: The paper exclusively investigates speech and music hallucinations, justifying this focus with dataset biases. However, it does not explore other plausible forms of acoustic hallucinations. For instance, a model might generate the sound of a drum being hit when a drum is merely visible in the scene, even if no one is striking it. Such audio hallucinations are equally likely to occur and can be just as detrimental as speech or music hallucinations. This possibility warrants careful verification to fully understand the model's failure modes.
* Concerns Regarding the Evaluation Metric: While the Majority Vote (MV) ensemble is employed, its precision is lower than that of a single model (PMNNs), and its recall is lower than two of the individual detectors. This raises the question of whether the ensemble approach is genuinely more effective. A more rigorous ablation study is needed to justify the choice of MV over potentially using a single, highly precise detector, as the current marginal benefit in robustness may not outweigh the performance trade-offs.
* Methodological Concerns in Hallucination Suppression: The observation that randomly masking video features already reduces the hallucination rate compared to the baseline model is intriguing and warrants deeper investigation. It prompts a critical question: does the proposed PFC method genuinely mitigate the root cause of hallucinations, or does it simply suppress diverse audio generation by selectively removing visual information? This casts doubt on whether the improvement is attributable to the precise detection of problematic segments or is merely a side effect of feature removal. The potential negative impact on the diversity and richness of the generated audio deserves further analysis.

**Questions:**

Please refer to the Weaknesses section.

**Details Of Ethics Concerns:**

No concerns.

---

### Official Review · Reviewer_aN2g · 2025-11-02

**Soundness:** 3
**Presentation:** 2
**Contribution:** 2
**Rating:** 4
**Confidence:** 4

**Summary:**

The paper addresses a critical, yet overlooked, failure mode in Video-to-Audio ($\text{V2A}$) generation models: generating sounds that do not correspond to the visible scene.

The paper introduces the concept of Insertion Hallucination ($\text{IH}$) in $\text{V2A}$ models, defining it as the spurious generation of structured acoustic events, specifically speech and music, that have no corresponding visual source. This failure mode is attributed to dataset bias where speech and music frequently occur off-screen in training data like $\text{VGGSound}$.To address this, the authors contribute:Formal Definition and Metrics: Formal definition of $\text{IH}$ and two new quantitative metrics: $\text{IH}@\text{vid}$ (hallucination prevalence per video) and $\text{IH}@\text{dur}$ (hallucinated duration severity).Detection Framework: A validated multi-detector ensemble (Majority Vote of $\text{inaSpeechSegmenter}$, $\text{YAMNet}$, and $\text{PANNs}$) to reliably detect $\text{IH}$.Mitigation Method (PFC): Posterior Feature Correction ($\text{PFC}$), a training-free, two-pass inference method that detects hallucinated segments and masks the corresponding video features with an empty representation in a second generation pass.

**Strengths:**

The definition of Insertion Hallucination ($\text{IH}$) as an auditory phenomenon (not a textual error, as in $\text{LLM}$s) in $\text{V2A}$ generation is a **fundamental and overlooked contribution**. The mitigation method, **Posterior Feature Correction ($\text{PFC}$)**, is highly original as a training-free, temporal-aware, self-correction mechanism for $\text{V2A}$ that intervenes directly at the feature level.

The work is grounded in meticulous validation. The detection framework is rigorously validated against a human-annotated dataset using the $F_{0.5}$-score (emphasizing precision/reliability). The ablations confirm that $\text{PFC}$'s success relies on precise temporal correction (outperforming random and complement replacement). The systematic testing across multiple benchmarks ($\text{Kling-Audio-Eval}$, $\text{VGGSound}$, $\text{AVE}$) confirms $\text{IH}$ as a widespread risk.

The paper clearly articulates the problem's cause ($\text{dataset bias}$ from off-screen sounds) and the proposed solution's mechanism ($\text{two-pass detection/correction}$). The metrics $\text{IH}@\text{vid}$ and $\text{IH}@\text{dur}$ are simple, intuitive, and precisely defined. Figure 3 clearly visualizes the $\text{PFC}$ process.

This work establishes authenticity/realism as a new, critical dimension for evaluating $\text{V2A}$ models, moving beyond simple temporal alignment and semantic similarity. The $\text{PFC}$ method offers an immediate, practical way to reduce $\text{IH}$ by over 50% on average across models without degrading standard quality metrics This paves the way for more reliable and trustworthy generative media models under the principle of "What You See Is What You Get".

**Weaknesses:**

Inference Latency of Two-Pass Process: The $\text{PFC}$ method requires two full generation passes through the $\text{V2A}$ model in addition to the detection time. This inherently doubles the inference latency, which is a significant drawback for real-time or production applications, especially for large models like $\text{MMAudio}$ and $\text{ThinkSound}$. Suggestion: A quantitative analysis of the added latency and a discussion on strategies to minimize the overhead (e.g., lightweight sampling in the first pass) is missing and crucial for practicality.

Sensitivity to Detector Error ($\text{False Negatives}$): $\text{PFC}$'s effectiveness is entirely predicated on the $\text{IH}$ detector's output. While the detector is validated for high precision ($F_{0.5}$ score), a False Negative (a true hallucination is missed) means the problematic features are retained, and $\text{IH}$ is not mitigated. A False Positive (a correct sound is flagged) means the features are erroneously masked, potentially deleting correct content. The former is a greater risk for the core mission. Suggestion: Discuss the trade-off of the $\beta$ value ($\beta=0.5$ favors precision) and its impact on the undetected IH rate ($\text{False Negatives}$).

Limitations of Masking with Empty Token: Replacing video features with a fixed empty token is a simple and effective hack. However, this token may represent a very specific "unconditional" state learned during training that might not generalize well across different $\text{V2A}$ architectures or even across different $\text{IH}$ types (speech vs. music). Suggestion: Discuss why the empty token is superior to alternative feature masking strategies, such as setting the features to zero, or using a mean/median feature vector.

**Questions:**

Quantitative Latency Analysis: Please provide a direct quantitative comparison of the inference latency (in seconds or ms) for a fixed-length clip (e.g., 5s) between the Baseline model and Baseline + PFC. Given the two-pass requirement, quantifying this trade-off is essential for assessing the method's practical utility.

Impact on Missing Non-IH Sounds: The core purpose of PFC is to prevent the insertion of IH. Does the intervention inadvertently make the model less likely to generate correct, but weak/ambiguous on-screen sounds? For instance, does removing the "unreliable visual features" at a specific moment suppress the generation of a quiet on-screen event that was correctly generated in the first pass? A discussion or a dedicated metric on this "suppression of correct events" would strengthen the analysis.

Generalization to New IH Types (Beyond Speech/Music): The authors focus on speech and music because they are the most frequent off-screen sounds and have mature detectors. However, IH is defined more broadly as the generation of any structured acoustic event without a visual source. Can the authors comment on extending this framework to other frequent off-screen categories found in VGGSound (e.g., Crowd Murmur) and the challenges associated with developing reliable detectors for these less-structured sounds?

Implications of KL and FD Improvements: Table 2 shows several instances where PFC leads to noticeable improvements in conventional metrics (FD and KL), such as a 10.3% gain in FD for ThinkSound on AVE. This suggests PFC is not merely non-destructive but can be constructive. Could the authors elaborate on the hypothesis that removing spurious auditory modes (IH) leads to a cleaner latent distribution, thereby improving distributional metrics?

---

### Note · Authors · 2025-11-14

I have read and agree with the venue's withdrawal policy on behalf of myself and my co-authors.